# Metastases to the Thyroid Gland: What Can We Do?

**DOI:** 10.3390/cancers14123017

**Published:** 2022-06-19

**Authors:** Qiushi Tang, Zhihong Wang

**Affiliations:** 1Chinese Journal of Practical Surgery, China Medical University, 9 Nanjing South St., Shenyang 110001, China; tangqiushi@vip.163.com; 2Department of Thyroid Surgery, The First Hospital of China Medical University, 155 Nanjing Bei Street, Shenyang 110001, China

**Keywords:** thyroid neoplasms, cancer metastasis, diagnosis, treatment

## Abstract

**Simple Summary:**

To better improve the clinical diagnosis and treatment of thyroid metastatic cancer, minimize morbidity and mortality, and improve the prognosis of patients, in this paper, we review the research status of thyroid metastatic cancer. Metastases to the thyroid gland itself are not common, and the clinical manifestations are not specific. Most of the primary tumors originate from the kidney, colorectum, lung, and breast. FNAB combined with IHC analysis is a specific method for the diagnosis of secondary thyroid neoplasms, and besides having high accuracy, it can also distinguish the primary location of the tumor. Surgical treatment is still the main treatment, supplemented by necessary radiotherapy and chemotherapy after surgery. The treatment plan for patients should be individualized and jointly developed by a multidisciplinary team. The prognosis of patients with thyroid metastatic cancer is still not optimistic. Therefore, we believe that when thyroid abnormalities are found in patients with previous or present malignant tumors, the possibility of metastases to the thyroid gland should be considered first.

**Abstract:**

Metastases to the thyroid gland arise from other malignant tumors such as renal cell carcinoma, colorectal cancer, lung cancer, and breast cancer. In clinical practice, the incidence is low, and the symptoms are not specific, so it is often missed and misdiagnosed. It is finally diagnosed via the comprehensive application of many diagnostic methods, such as ultrasound, fine-needle aspiration biopsy, and immunohistochemistry analysis. Surgery-based comprehensive treatment is often adopted, but because it is usually in the late stage of the primary tumor, the prognosis is poor. In order to better understand the related characteristics of thyroid metastatic cancer and then improve the clinical diagnosis and treatment and the prognosis of patients, in this paper, we systematically summarize the research status of thyroid metastatic cancer.

## 1. Introduction

Metastases to the thyroid gland are a kind of disease that metastasizes from nonthyroid malignant tumors to the thyroid gland. It is clinically rare, with an incidence of 0.36% in all thyroid malignant tumors [1], but is more common in autopsy reports, with an incidence of approximately 1.9% to 24% [2,3,4,5]. This incidence shows that thyroid metastatic cancer is often missed and misdiagnosed in the clinic. The average age of patients with thyroid metastatic cancer is 60 to 70 years old, and most of them are female [5,6,7]. Since it is in the late stage of the primary tumor, the prognosis of thyroid metastatic cancer is very poor and is closely related to the congenital characteristics of the primary tumor. Usually, the average survival time of patients after thyroidectomy is 43.2 months [6,8]. In addition, the factors related to the prognosis of thyroid metastatic cancer include the time, quantity, size, and location of metastasis at the time of diagnosis, the methods of treatment, and the health status of the patients [9]. Russell et al. found that the longer survival time of patients is related to the long interval between the diagnosis of the primary tumor and the occurrence of metastases to the thyroid gland. Thus, when the primary tumor has stronger inertia (such as kidney and breast tumors), the survival rate of patients is higher [10]. More importantly, approximately 35% to 80% of thyroid metastatic cancer patients have multiple organ metastasis, which leads to a poor prognosis [8,11,12]. Patients with slow primary lesions and limited metastasis to the thyroid gland without obvious extrathyroid invasion and metastasis had a relatively better prognosis [9]. Interestingly, a study found that black race, older age of patients at diagnosis of the primary tumor, male sex, and three or more primary tumors all contributed to lower overall survival in patients with metastases to the thyroid gland, among which black race and older age of primary tumor were the most significant independent risk factors for overall survival in these patients [13]. In the past decade, with progress in the diagnosis and treatment of cancer patients and the improvement of the patient survival rate, the number of thyroid metastatic cancers has been increasing [14,15]. Therefore, to better improve the clinical diagnosis and treatment of thyroid metastatic cancer, minimize morbidity and mortality, and improve the prognosis of patients, in this paper, the research status of thyroid metastatic cancer is reviewed. The PubMed and Web of Science databases were searched for publications with the keywords “metastasis to thyroid gland” or “secondary thyroid neoplasms”. The search was conducted both on basis of the MESH tree and as a text search. Inclusion criteria were research studies in English, peer-reviewed, centered on metastasis to the thyroid gland, published between January 1979 and December 2021. Exclusion criteria were systematic reviews, studies focused on metastasis of thyroid cancer, or articles published in languages other than English. 

## 2. The Cause and Mechanism of the Metastasis

The method of transfer can be divided into synchronous metastasis and metachronous metastasis. The former refers to the simultaneous detection of the primary tumor and secondary thyroid neoplasms, while the latter refers to the discovery of secondary thyroid neoplasms 6 months after the diagnosis of the primary tumor [4,10]. Generally, the average interval between the discovery of the primary tumor and the occurrence of metastases to the thyroid gland is 3.3 years. Depending on the characteristics of the primary tumor, when it is more aggressive, the interval is shorter [16]. Metachronous metastasis (approximately 69%) is the main method of transfer of secondary thyroid neoplasms [4], and its long interval may lead to missed diagnosis and misdiagnosis of secondary thyroid neoplasms. There are three main metastatic pathways of metastases to the thyroid gland: direct metastasis, such as for esophageal, laryngeal, and pharyngeal tumors; retrograde lymph node metastasis, such as for lung and breast tumors; and hematogenous metastasis, such as for kidney and gastrointestinal tumors [17]. However, because the thyroid gland is a hypervascular organ, most metastases to the thyroid gland are hematogenous [18].

In terms of metastatic mechanism, as early as 1931, Willis proposed two hypotheses to explain the low incidence of secondary thyroid neoplasms: on the one hand, the rapid arterial blood flow of the thyroid inhibited the adhesion of malignant tumor cells; on the other hand, high oxygen saturation and high iodine concentration in the thyroid inhibited the growth of malignant tumor cells [19]. At present, it is also believed that the abundant blood supply of the thyroid gland can inhibit the deposition of tumor thrombus. When thyroid lesions occur, poor or interrupted blood supply is beneficial to the deposition of tumor cells [20], and the concentration of iodine and oxygen in abnormal thyroid is decreased, so metastases to the thyroid gland are more likely to occur [21,22]. Studies have found that a multinodular goiter or pathological state of the thyroid can induce tumor cells to be implanted in other organs [6]. However, a recent study suggests that the abundant blood supply of the thyroid is the reason why it has become a target for cancer metastasis [23].

## 3. The Primary Tumor

Global studies have shown that the most common primary tumors of secondary thyroid neoplasms in autopsy originate from the lung, breast, kidney, and colorectal cancer, while in the clinic, renal cell carcinoma is the most common, followed by colorectal cancer and lung cancer, which may be due to the more aggressive nature of lung malignant tumors [1,5,9]. Domestic studies have shown that metastases to the thyroid gland are most commonly derived from lung cancer, followed by breast cancer and gastric cancer [16,24]. This difference is probably caused by the difference in tumor incidence and primary tumor invasiveness at home and abroad. Histologically, adenocarcinoma and squamous cell carcinoma are the main histological types of primary tumors [5]. A recent review estimated the frequency of metastasis of related malignant tumors to the thyroid gland to be 48% in renal cell carcinomas, 10% in colorectal cancers, 8% in lung cancers, and 8% in breast cancer [23] (Figure 1).

### 3.1. Renal Cell Carcinoma (RCC)

RCC is the most common malignant tumor metastatic to the thyroid gland. RCC metastasized to the thyroid gland is described as a clear cell type [7,25]. The median time of thyroid metastasis in patients with RCC was 92 months, and the median survival time after thyroid metastasis was 54 months [26]. Previous studies have suggested that the average incubation period of metastasis of renal clear cell carcinoma to the thyroid gland is 8.7 years [7]. Patients with RCC metastasis to the thyroid can be symptomatic, have asymptomatic accidental discoveries, or have simple painless masses. Generally, the size of metastatic foci in symptomatic patients is larger than that in simple painless masses and asymptomatic patients, and there are few symptoms, such as dysphagia or dyspnea [7,27]. To avoid further progression of the disease to the central neck, total thyroidectomy should be considered [26]. Beutner et al. found that, after active surgical resection, the median survival time of patients with metastatic RCC to the thyroid gland was 6.5 years, while that of patients with thyroid metastasis of other primary tumors was only 4.7 years [28]. These findings may be related to the weak invasiveness and malignant degree of RCC. Therefore, the European Association of Urology guidelines supports the surgical treatment of metastases to the thyroid gland [29]. In addition to surgical treatment, sunitinib is also an effective drug for the treatment of metastatic RCC [27]. However, because RCC itself has a certain resistance to radiotherapy, postoperative radiotherapy is not recommended [30]. As RCC patients still have severely delayed metastasis after nephrectomy, RCC patients should always be on guard against the occurrence of RCC thyroid metastasis, even a few years after nephrectomy [31].

### 3.2. Lung Cancer

Studies have shown that nonsmall cell lung cancer is the most common type of lung cancer metastatic to the thyroid gland [32]. Adenocarcinoma is the most common form of nonsmall cell lung cancer, but there have also been reported cases of small cell lung cancer and squamous cell carcinoma metastasizing to the thyroid gland [33]. The chief complaints of many patients with thyroid metastasis of lung cancer are nonspecific, such as dyspnea, dysphagia, fatigue, and cough [33]. The study found that older women who do not smoke or Asians who smoke less are more likely to get sick, which may be due to genetic factors [34]. EGFR and EML4-ALK gene mutations are associated with thyroid metastasis of lung cancer, but the exact mechanism is not clear [33]. The combined use of morphological fine-needle aspiration biopsy (FNAB) features with immunohistochemistry (IHC) analysis of adequate cell content on smeared specimens, as well as on cell-block material, is a feasible and reliable method to achieve a definitive diagnosis of metastatic nonsmall cell lung cancer to the thyroid gland [35]. In terms of prognosis, the prognosis of patients with adenocarcinoma was significantly better than that of patients with squamous cell carcinoma or other histological types [36].

### 3.3. Colorectal Cancer (CRC)

The average age of CRC patients with thyroid metastasis was 61 years, which was similar to that of CRC patients, and the average interval between the diagnosis of primary tumor and metastases to the thyroid gland was 51 ±31 months, while the average survival time of CRC patients with thyroid metastasis was only 11.3 months [37]. FNAB usually shows two different types of cells: thyroid and colorectal [38,39]. Kumamoto et al. found that the most common site of origin (excluding the rectum) in patients with thyroid metastasis of CRC was the right ascending colon [40]. Since metastases to the thyroid gland in most CRC patients occur after lung metastasis [41], any patient with a history of primary CRC (especially if there is also a history of lung metastasis) should consider the possibility of metastases to the thyroid gland [42].

In addition to the above three common clinical primary tumors, melanoma is also associated with metastases to the thyroid gland, and autopsy reports show that the incidence is as high as 39% [8,43]. Overall, for metastases to the thyroid gland with metachronous metastasis, the average interval between the diagnosis of the primary tumor and the occurrence of metastases to the thyroid gland was 6.8 years [44]. Secondary thyroid neoplasms were found in patients with primary tumors at an average of 39.5 months after the operation, and this interval was the longest in patients with RCC and the shortest in patients with lung cancer [17].

## 4. Diagnosis

For patients with a clear history of malignant tumors, the possibility of secondary thyroid neoplasms should be considered when there are clinical manifestations of thyroid abnormalities. Common examination methods for patients with metastases to the thyroid gland include ultrasound, FNAB, IHC analysis, and positron emission tomography–computed tomography (PET–CT), which are usually performed when new palpable nodules or goiters appear in the neck of the patient.

### 4.1. Clinical Manifestation

The main clinical manifestations of the patients were goiters (painless nodules that can move up and down with swallowing), neck swelling, dysphagia, dysphonia, hoarseness, and cough similar to primary thyroid tumors [10,17]. These symptoms are closely related to the primary location of the tumor. Tumors originating from adjacent organs (such as the trachea, esophagus, or larynx) usually cause symptoms such as dyspnea, dysphagia, or hoarseness [16]. As the thyroid function of most patients with metastases to the thyroid gland has not been timely and accurately recorded, there are few clinical data available for reference. Hypothyroidism was previously found in patients with thyroid metastasis of lung cancer because follicular cells were replaced by cells from the lungs, which infiltrated into the thyroid gland, resulting in a decrease in follicular cells [45]. However, some recent studies have pointed out that the thyroid function of most patients with metastases to the thyroid gland is normal, but some patients may have hypothyroidism or hyperthyroidism [14,46]. It has been found that patients with hyperthyroidism have similar underlying pathogenesis to subacute thyroiditis, that is, the damage or destruction of thyroid tissue, which leads to the uncontrolled release of thyroid hormones (mainly thyroxine) into circulation. However, with the treatment and control of the primary tumor, thyroid function is likely to recover; nevertheless, if the primary tumor continues to progress, hypothyroidism is likely to eventually occur [46]. In terms of clinical manifestations, the prominent feature of patients with thyroid function changes is painful goiter, sometimes accompanied by oppressive symptoms, and the degree of hyperthyroidism varies from subclinical to severe [46].

### 4.2. Ultrasound

Ultrasound is the first choice for imaging examination of thyroid diseases. In recent years, with the widespread application of ultrasound, the detection of metastases to the thyroid gland has improved [47,48]. The typical ultrasonographic findings of secondary thyroid neoplasms include ill-defined hypoechoic nodules and intranodular angiogenesis [9], and the size of multiple nodules is between 0.6 and 6.6 cm [16]. Saito et al. divided secondary thyroid neoplasms into two types according to their ultrasonographic findings: diffuse type, showing diffuse hypoechoic lesions involving the whole thyroid, and nodular type, showing hypoechoic nodular lesions in the thyroid with a low degree of vascularization [49]. Debnam et al. [50] found that the ultrasonographic findings of secondary thyroid neoplasms were similar to those of benign and malignant thyroid diseases and were mainly divided into two types—namely, 65% of patients showed solitary or partially multinodular thyroid nodules, and 35% of patients showed diffuse infiltrative disease. The former is mainly characterized by solid nodules, hypoechoic margins, ill-defined margins, no hypoechoic halo, punctate hyperechoic foci, and vascularity of varying degrees; on the other hand, the characteristics of the latter are not specific and cannot be distinguished from thyroiditis or other diffuse diseases in the thyroid gland. All these results are consistent with the results of previous studies by Saito et al. In addition, in some cases, cystic echo can be detected by ultrasound, which may be due to the rapid growth of metastatic cancer, insufficient blood supply, followed by necrosis and liquefaction, to form cystic degeneration [51]. Moreover, in a previous study, Gregorio et al. further indicated that ultrasounds provided by endocrinologists and surgeons were more accurate in predicting malignancy, compared with those provided by radiologists [52].

### 4.3. PET–CT

Compared with traditional CT, which only focuses on the macroscopic anatomical structure, PET–CT is more aimed at microscopic cells and molecules, so PET–CT is more sensitive to the diagnosis of tumors. More importantly, because PET–CT involves whole-body imaging, it has incomparable advantages in finding the primary focus of the disease or monitoring tumor recurrence or metastasis. Therefore, PET–CT is recommended as one of the items for long-term follow-up of patients with malignant tumors and can be helpful in detecting metastases to the thyroid gland in time [53]. Metastases to the thyroid gland showed focal single nodular uptake, multiple discrete nodular uptakes, or diffuse uptake/infiltration on 18F-PET–CT, and the standardized uptake value (SUV) ranged between 3.9 and 42 [54]. Therefore, for patients with previous or present extrathyroid malignant tumors, potential metastases to the thyroid gland should be suspected when there is a thyroid mass or diffuse infiltration on 18F-PET–CT [54].

### 4.4. FNAB

If the patient has a history of other malignant tumors and is suspected to have secondary thyroid neoplasms, FNAB may obtain a correct diagnosis [4], and useless thyroidectomy can be avoided for patients with poor prognosis [30,55]. A multi-institutional study on FNAB showed that 87% of patients can be diagnosed with thyroid malignant tumors by FNAB, and 93% of them can be specifically diagnosed with metastases to the thyroid gland [56]. However, the correct rate of FNAB diagnosis depends on the location of the primary tumor; the correct rate of diagnosis is the lowest when the primary tumor is esophageal cancer, which is only approximately 50% of the time, and the correct rate is the highest when the primary tumor is breast cancer [6]. The positive and negative predictive values of FNAB in metastases to the thyroid gland were 89% and 93%, respectively. The application of molecular markers (e.g., the application of BRAF growth promoter in thyroid papillary carcinoma) and IHC (e.g., the application of CD-10 in RCC) may be further helpful in the diagnosis of FNAB [57,58,59,60]. Therefore, patients with previous or present extrathyroid malignant tumors should be regarded as having potential intrathyroidal metastasis when thyroid nodules or diffuse thyroid infiltration are found via ultrasound. Regardless of where the primary tumor is located, it should be evaluated using ultrasound-guided FNAB [50]. However, although secondary thyroid neoplasms retain the histological features of primary tumors, pathological diagnosis may be difficult because they are usually poorly differentiated [5]. The main problem is that it is difficult to distinguish primary thyroid tumors from secondary thyroid neoplasms.

### 4.5. IHC Analysis

IHC analysis is helpful in distinguishing secondary thyroid neoplasms from primary thyroid tumors; for example, positive immunological staining of thyroglobulin and thyroid transcription factor-1 (TTF-1) indicates primary thyroid tumors, while secondary thyroid neoplasms usually do not express these markers but express specific markers related to their origin tissues [5,25,61,62,63]. Therefore, IHC analysis can also be used to determine the source of primary tumors in metastases to the thyroid gland. Among them, the IHC analysis of thyroid follicular cell markers is the most helpful in distinguishing primary thyroid tumors from secondary thyroid neoplasms [1]. PAX2 and/or PAX8, CD10, and RCC antibodies are generally considered to be the most useful markers for the diagnosis of thyroid metastasis and nephrogenic metastasis [4,64]. When it is highly suspected that metastases to the thyroid gland originate from the lung, monoclonal PAX8 can be used to assist in the diagnosis [1]. Among primary tumors that metastasize to the thyroid gland, breast cancer is one of the most common. GATA-3, estrogen, and progesterone receptors are helpful in diagnosing the origin of breast cancer [4,65]. CRC cells are positive for CK20 and CEA but negative for thyroglobulin, calcitonin, and TTF-1 [38,39], which is helpful for the diagnosis of metastases to the thyroid gland of CRC origin. In addition, in an autopsy study on secondary thyroid tumors released in 2019, the immunohistochemical results of the above four most common primary tumors were reviewed and analyzed in detail [5] (Table 1).

Rare, and hence morphologically unfamiliar, tumors and lack of cellular differentiation and expression of antigens conveniently detected via IHC are considered to be some of the difficulties [22,66].

### 4.6. Histopathology

The morphological evaluation of hematoxylin and eosin-stained sections is very helpful in the diagnosis of secondary thyroid neoplasms [5]. The cytological findings of secondary thyroid neoplasms mainly depend on the type of primary tumor and the degree of thyroid involvement. Focal metastatic lesions may be a mixture of malignant cells and normal follicular cells, while diffuse metastases to the thyroid gland may be cytological samples composed entirely of malignant cells [67]. Therefore, when atypical cells that do not meet the criteria of primary thyroid tumors are detected, the possibility of metastases to the thyroid gland should be suspected [67]. However, the most difficult aspect of morphological diagnosis is thyroid metastasis of RCC and breast cancer because these tumors may show an alveolar/glandular structure resembling a follicular pattern made up of clear cells, which is not uncommon in thyroid hyperplastic nodules [4]. Therefore, to make a better and more accurate diagnosis of secondary thyroid neoplasms, it is necessary to be familiar with the cytological characteristics of common primary tumors related to secondary thyroid neoplasms [68].

### 4.7. Differential Diagnosis

Since the clinical manifestation of metastases to the thyroid gland is hidden and lacks specificity, it is necessary to distinguish it from primary thyroid tumor, benign thyroid nodule, local invasive disease of the larynx or esophagus infiltrating the thyroid gland, and other diseases. The most important aspect is to distinguish it from the primary thyroid tumor. Unlike primary thyroid tumors, metastases to thyroid gland tumor cells show interstitial infiltration and metastasis, resulting in follicles surrounded by tumor cells and leading to deformities, but follicles are rarely infiltrated by tumor cells [5]. These, together with the IHC and cell morphology mentioned above, are helpful in distinguishing primary thyroid tumors from metastases to the thyroid gland.

## 5. Treatment

### 5.1. Treatment Method

The treatment of secondary thyroid neoplasms mainly includes surgery, radiotherapy, and chemotherapy. At present, surgery is the main comprehensive treatment, including total thyroidectomy or subtotal thyroidectomy plus cervical lymph node dissection, postoperative chemotherapy or radiotherapy, and long-term oral thyroxine tablets to replace thyroid function and inhibit pituitary thyrotropin secretion to reduce the recurrence of cancer [33]. Thyroidectomy is considered to be a more effective treatment than radiotherapy and chemotherapy [4]. However, it is still controversial whether patients with metastases to the thyroid gland must be treated by surgery. Generally, patients with extensive metastasis of malignant tumors rarely have surgical indications, but when they only metastasis to the thyroid gland, surgery may achieve cancer control or even a long-term cure. In this case, surgical treatment should be fully considered [6,9]. In addition, thyroidectomy is also recommended when the thyroid gland is the only organ involved in metastases, secondary to differentiated tumors, and the patient has a good life expectancy [17]. Russel et al. pointed out that patients who have undergone thyroidectomy survive longer, especially those with metastatic RCC [10], which may be related to the weak invasiveness of RCC. Although surgery is related to improving the survival rate of patients, the overall survival rate of patients with thyroid metastasis of CRC and lung cancer is lower because of the stronger invasiveness of their primary tumors [17]. Therefore, some people believe that thyroidectomy is not recommended for highly invasive primary tumors such as lung cancer because of their poor prognosis [16]. It should be added that when patients with metastases to the thyroid gland develop oppressive symptoms, thyroidectomy [4] can also be considered to relieve the symptoms and improve the quality of life of the patients. Compared with primary thyroid tumors, secondary thyroid neoplasms are not sensitive to radioactive iodine [9]. Therefore, for patients who cannot undergo surgical treatment, radiotherapy or chemotherapy is feasible, but the effect is not ideal [6,10]. This is partly because RCC is the most common primary tumor, which easily metastases to the thyroid gland, and it is considered to be resistant to radiotherapy to a large extent [69]. Recent studies have shown that immune checkpoint inhibitors and tyrosine kinase inhibitors may have greater application prospects in metastatic RCC than in traditional palliative surgery [7].

### 5.2. Surgical Strategy

At present, there is still much controversy about the scope of resection of secondary thyroid neoplasms. In unilateral disease, most authors recommend thyroid lobectomy rather than total thyroidectomy to minimize the risk to the contralateral recurrent laryngeal nerve and parathyroid glands [9]. However, some authors suggest that lobectomy can be associated with positive margins and, therefore, favor total thyroidectomy [70]. Russell et al. demonstrated a decrease in recurrence for patients managed with total thyroidectomy versus thyroid lobectomy, but there was no significant difference in the incidence of operative complications between the two groups [10]. Iesalnieks et al. pointed out that, for patients with multiple, small, and bilateral lesions, total thyroidectomy should always be performed to avoid recurrence [12]. Conversely, another review report revealed that patients with negative incisal margins had no tumor recurrence and that total thyroidectomy did not bring a better survival rate than thyroid lobectomy [28]. However, recent studies still believe that total thyroidectomy is preferable [17]. These findings notwithstanding, a negative incisal margin is a prerequisite to ensure that cancer does not recur in patients with metastases to the thyroid gland. In addition, since regional lymph node involvement is not common, preventive cervical lymph node dissection is generally not recommended [9]. However, regional lymphatic metastasis should be comprehensively evaluated before the operation, especially if the primary tumor is RCC [71]. In terms of prevention, Pitale et al. suggest that when screening cancer patients for chest CT, thyroid-chest CT should be used to avoid missing metastases to the thyroid gland [72].

In short, when treating patients with metastases to the thyroid gland, we should comprehensively consider the characteristics and metastasis of the primary tumor, as well as the patient’s tolerance to surgery and life expectancy, and adopt an individualized treatment plan with multidisciplinary cooperation.

## 6. Issues and Prospects

Although there have been an increasing number of studies about metastases to the thyroid gland in recent years, there are still many shortcomings and limitations. First, as mentioned above, a dysfunctional thyroid is likely to promote the occurrence and development of metastases to the thyroid gland [6,20,21,22]. Therefore, can thyroid dysfunction be used as an independent risk factor for predicting the occurrence of metastases to the thyroid gland in patients with a clear history of malignant tumors? At present, there is no effective evidence to prove it, so in future research, we can focus on whether there is a significant statistical correlation between “thyroid dysfunction” and “incidence of metastases to the thyroid gland”. In this process, it should be noted that the event of “thyroid dysfunction” occurred before the occurrence of metastases to the thyroid gland, and thus, the thyroid dysfunction was not caused by metastases to the thyroid gland. Attention should be given to whether the microenvironment composed of high oxygen and iodine content in the thyroid plays a certain role in the occurrence and development of metastases to the thyroid gland. In terms of demographic data of patients, some studies have found that most patients are female [6], while other studies have found that the proportion of male patients is slightly higher than that of female patients [5]. However, a recent study found that the male-to-female ratio of patients with secondary thyroid neoplasms was 1:1 [1]. Therefore, there is still no strong evidence to prove that sex plays an important role in the occurrence and development of secondary thyroid neoplasms [8]. In addition, most of the literature about secondary thyroid neoplasms has not mentioned the level of TSH in patients, and there is no direct evidence to prove that the level of TSH is related to the occurrence and development of secondary thyroid neoplasms. In addition, are there other metastatic foci in patients with secondary thyroid neoplasms in addition to the thyroid gland? For example, Iesalnieks et al. and Beutner et al. found simultaneous pancreatic metastasis in 14 patients (31.1%) and 11 patients (32.4%), respectively, in their case series of RCC thyroid metastasis [12,28]. If so, is there any connection between these metastatic foci, and what is the correlation between them?

Second, in terms of diagnosis, FNAB cannot guarantee a clear diagnosis of all secondary thyroid neoplasms. The most common primary tumors without a correct diagnosis via FNAB are esophageal, cervical, RCC, and malignant melanoma, while the primary tumors with the highest correct diagnosis rate of FNAB are breast cancer, lung cancer, and CRC [17]. Although FNAB is a sensitive and effective tool for the diagnosis of secondary thyroid neoplasms, it is sometimes unable to make an accurate diagnosis, especially for patients with multiple histological origins and unclear histories of primary tumors. Moreover, when goiter or nodules of the thyroid gland appear in patients with extensive cancer, it is difficult to judge whether they are caused by the metastasis of the primary tumor to the thyroid gland, which is a difficult problem to diagnose even at autopsy [5]. PET–CT is an effective method for the diagnosis of cancer metastasis. In addition to the timely detection of cancer metastasis to the thyroid gland in the diagnosis of secondary thyroid neoplasms, there is still a lack of relevant research on the specific location and staging of the primary tumor. For example, can an SUV be used to accurately predict and diagnose the characteristics of primary tumors? Moreover, when secondary thyroid neoplasms show single or multiple nodules on PET–CT, it is difficult to distinguish them from single nodular uptake in primary thyroid tumors [54]. At present, research on the diagnosis of metastases to the thyroid gland is mainly focused on ultrasound, FNAB, IHC, etc., but there is a lack of research on molecular detection. A recent article pointed out that molecular detection of patients may show characteristic changes in the primary tumor, which is helpful in identifying the primary location of the tumor without identifying the primary tumor and existing extensive metastasis [1]. Therefore, in the next study, we will perform a comprehensive and systematic analysis of the metastases to the thyroid gland of different primary tumors at the molecular level. This is not simply conducive to accurately identifying the primary site, but more importantly, it can help people to better understand the pathogenesis of metastases to the thyroid gland and then find targeted therapeutic drugs for specific mutant genes.

Finally, when it is decided to perform surgery on patients with metastases to the thyroid gland, the scope of surgery is also a controversial issue that needs to be solved urgently; that is, there is a lack of unified guidelines for the surgical treatment of metastases to the thyroid gland. With regard to the prognosis of patients with metastases to the thyroid gland, the existing studies have mainly focused on the characteristics of the primary tumor and the treatment measures taken for the patients, but the role of genetic and biological factors in this process has largely been ignored. This provides a new idea for future research, that is, to explore the role of genetic and biological factors in the occurrence and development of secondary thyroid neoplasms. In addition, it should be added that there may be deviations and limitations in some conclusions due to differences in data sources, sample size, data quality, and data processing methods in different studies in the literature.

## 7. Conclusions

In summary, metastases to the thyroid gland itself are not common, and the clinical manifestations are not specific. Most of the primary tumors originate from the kidney, colorectum, lung, and breast. FNAB combined with IHC analysis is a specific method for the diagnosis of secondary thyroid neoplasms, and in addition to having high accuracy, it can also distinguish the primary location of the tumor. Surgical treatment is still the main treatment, supplemented by necessary radiotherapy and chemotherapy after surgery. The treatment plan for patients should be individualized and jointly developed by a multidisciplinary team. However, the prognosis of patients with thyroid metastatic cancer is still not optimistic. Therefore, when thyroid abnormalities are found in patients with previous or present malignant tumors, the possibility of metastases to the thyroid gland should be considered first.

## Figures and Tables

**Figure 1 cancers-14-03017-f001:**
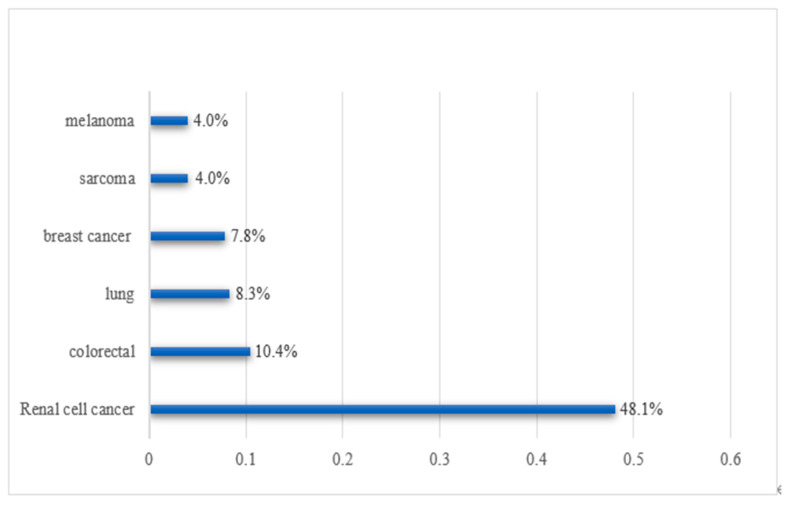
Frequency of metastasis of related malignant tumors to the thyroid gland.

**Table 1 cancers-14-03017-t001:** Histological and immunohistochemical characteristics of secondary tumors of the thyroid gland.

Primaries	Number of Cases (*n*)	Subtypes Cases (*n*)	Immunohistochemistry
Lung	12	Adenocarcinoma (6)	Tg (−), Ck7(+), CEA (+), TTF (+)
		Squamous cell (4)	Tg (−), Ck 5/6 (+), p63 (+), TTF−1 (−)
		Large cell (1)	Tg (−), CEA (−), NSE (+)
		Small cell (1)	Tg (−), Chr (+), Ck20 (−), CEA (−)
Kidney	3	Clear cell carcinoma (3)	Tg (−), CEA (+), WT1 (+)
Breast	3	Ductal (2)	Tg (−), Her2 (+)
		Lobular (1)	Tg (−), Her2 (+)
Colorectal	1	Adenocarcinoma (1)	Tg (−), CEA (+), Ck20 (+)

Tg: thyroglobulin mAb, Ck: cytokeratin, CEA: carcinoembryonic antigen, TTF-1: thyroid transcription factor-1, NSE: neuron-specific enolase, Chr: chromogranin, WT1: Wilms’ tumor gene, Her2: human epidermal growth factor receptor-2, (+): positive immunostaining, (−): no immunostaining.

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
