# Peer review of "Metastases to the Thyroid Gland: What Can We Do?"

_cancers, 2022, doi:10.3390/cancers14123017_

Round 1

Reviewer 1 Report

This is a very interesting and comprehensive review concerning metastatic thyroid cancer. The overall approach of the manuscript can be considered original, the various paragraphs are developed organically and with considerable coherence, and the conclusions are interesting, considering that specific studies on this topic have not been sufficiently developed at present.

The references in support of what has been stated are in any case adequate.

Suggestions that can be made to improve the reading of the article are as follows

ABSTRACT

1)Metastases to the thyroid gland are caused by metastasis.....to the thyroid gland" could be changed with: "Metastases to the thyroid gland arise from other malignant tumours...to the thyroid gland" (delete "to the thyroid gland")

2) Surgery-based comprehensive treatment is often adopted in the treatment (delete)

3) "... improve the clinical diagnosis and treatment and improve the prognosis of patients..." (delete)

2nd PARAGRAPH

1) "Depending on the characteristics of the primary tumor, when the primary tumor is more aggressive" change with "when it is more aggressive.

2) "...most metastases to the thyroid gland are hematogenous metastases" (delete).

4th PARAGRAPH

1) section 4.1: Clinical manifestation, line 179: the reference number [45] should be moved at the end of the sentence, after the dot.

2) section 4.2. This section should also discuss "how and by whom" thyroid ultrasound should be interpreted, adding related citations. I might suggest, for example, the following article: doi: 10.1016/j.jss.2020.12.009.

3) section 4.3, line 221: "... single nodular uptake of the thyroid gland" - erase.

4) section 4.4: ...as an important tool for the diagnosis of secondary thyroid neoplasms: erase

5) The section concerning diagnostic cytopathology (4.4) should be enriched with a comment concerning the investigation of nodules originating from thyroid follicular cells. In particular, the Bethesda classification and its most recent amendments should be mentioned, also highlighting possible confounding factors that could classify some thyroid metastases as indeterminate or suspicious, and also "tumour-to-tumour" metastases, adding the relevant bibliographic data. Correctly, the authors conclude by emphasising the difficulty of distinguishing primary tumours from metastases, and a detailed discussion of this could make the criteria clearer.

Section 4.5, line 260: change "are pulmonary metastases" with "originate from lung".

Section 5.1, lines 329-330: the sentence "This is partly because the most common 329 primary tumor of metastases to the thyroid gland is RCC" should be rephrased because it is not very comprehensible.

Section 5.2: As is widely practised everywhere in the world, the choice between total thyroidectomy and lobectomy should also be made taking into account nerve monitoring data (add some references)

Author Response

Dear reviewer:

Thank you very much for taking the time to read and modify the article. And thank you for your valuable suggestions. I’ve studied the comments and carefully revised the manuscript item by item. please find the attached file as my reply letter.

Reviewer 2 Report

The work of Tang and Wang is a systematic review of the literature entitled “Metastases to the thyroid gland: What can we do?” The authors’ objective is to summarize previously published research to improve the clinical diagnosis and treatment, minimize the morbidity and mortality, emphasize importance of early detection and subsequently improve the prognosis of patients with neoplastic metastases to the thyroid gland.

Please specify search strategy, inclusion or exclusion criteria used for this review and how studies were grouped for the syntheses. Please add to the manuscript.

Please specify all databases, registers, websites, organizations, reference lists and other sources searched or consulted to identify studies.

Neoplastic metastases to the thyroid gland or metastases to the thyroid gland or metastatic cancer to the thyroid is very different from thyroid metastatic cancer and is different from thyroid cancer as second malignant neoplasm. Please explain.

Line 53 and 55 mention older age of primary tumor. What does the author intended to report?  Is it age of the tumor as how long after diagnosis of the tumor, or the patient’s age at diagnosis? Please clarify.

Line 56 … risk factors for death in these patients.  From line 52 to line 56, I suggest sentence review. Please explain.

Line 103 refers to Figure 1 and the only figure in this paper is labeled Figure 2. The figure is representative of results from another paper and not from the author’s synthesized conclusion. I suggest more figures including comparison of results within cited work.

Line 261 Breast cancer is one of the most common primary tumor in metastases to the thyroid gland. Please clarify.

Line 273 Please add Her2 to the legend of Table 1.

Line 312 to 316 has a redundant statement. Please review.

Overall, I do not see numerical or statistical analysis summary from previously published data cited by the authors, making it hard to conclude scientific significance, validate hypothesis or conclusion. Please include statistics relevant to conclusions.

The authors state that at present, research on the diagnosis of metastases to the thyroid gland lack on molecular detection. The authors are planning to perform a comprehensive and systematic analysis of the metastases to the thyroid gland of different primary tumors at the molecular level in the future without any other background information on molecular markers except for BRAF. Please explain.

Author Response

Dear reviewer:

Thank you very much for taking the time to read and modify my article. And thank you for your valuable suggestions. I’ve studied the comments and carefully revised the manuscript item by item. Please find the attached file as my reply letter.

sincerely yours,

Dr. Qiushi Tang
Prof. Zhihong Wang 
